# Relationship between Olfactory Function and BMI in Normal Weight Healthy Subjects and Patients with Overweight or Obesity

**DOI:** 10.3390/nu14061262

**Published:** 2022-03-16

**Authors:** Fernanda Velluzzi, Andrea Deledda, Maurizio Onida, Andrea Loviselli, Roberto Crnjar, Giorgia Sollai

**Affiliations:** 1Obesity Center, Department of Medical Sciences and Public Health, University of Cagliari, Presidio Ospedaliero San Giovanni di Dio, 09124 Cagliari, Italy; fernandavelluzzi@gmail.com (F.V.); andredele@tiscali.it (A.D.); onida.maur@gmail.com (M.O.); alovise2@gmail.com (A.L.); 2Department of Biomedical Sciences, University of Cagliari, Monserrato, 09042 Cagliari, Italy; crnjar@unica.it

**Keywords:** smell, olfactory dysfunction, obesity, body mass index (BMI), nutrition, gender

## Abstract

Smell plays a critical role in food choice and intake by influencing energy balance and body weight. Malnutrition problems or modified eating behaviors have been associated with olfactory impairment or loss. The obesity epidemic is a serious health problem associated with an increased risk of mortality and major physical comorbidities. The etiopathogenesis of obesity is complex and multifactorial, and one of the main factors contributing to the rapid increase in its incidence is the environment in which we live, which encourages the overconsumption of foods rich in energy, such as saturated fats and sugars. By means of the “Sniffin’ Sticks” test, we measured the olfactory threshold, discrimination and identification score (TDI score) in patients of the Obesity Center of the University Hospital (OC; *n* = 70) and we compared them with that of healthy normal weight controls (HC; *n* = 65). OC patients demonstrated a significantly lower olfactory function than HC subjects both general and specific for the ability to discriminate and identify odors, even when they were considered separately as females and males. For OC patients, a negative correlation was found between body mass index (BMI) and olfactory scores obtained by each subject, both when they were divided according to gender and when they were considered all together. Besides, normosmic OC patients showed a significantly lower BMI than hyposmic ones. A reduced sense of smell may contribute to obesity involving the responses of the cephalic phase, with a delay in the achievement of satiety and an excessive intake of high-energy foods and drinks.

## 1. Introduction

Several studies on the physiological processes involved in nutrition have shown that food intake depends on multiple and complex mechanisms involving both the central and peripheral nervous systems [1,2]. In particular, a relationship between olfactory function and eating behavior has been reported [3]. On the one hand, metabolic imbalances can change the olfactory function, while on the other, the olfactory function can affect the energy balance and body weight [3]. One of the main functions of the olfactory system is to direct food choices, playing an important role in the intake of nutrients and in the control of meal size [4,5,6]. By sending information from the external environment to the central nervous system, the sense of smell participates in the cephalic phase responses (CPR), that is, in those physiological adaptations that allow animals to quickly face the metabolic challenges related to food intake. Nevertheless, CPRs are not only used to prepare the body to digest, absorb and metabolize food, but also to start and finish a meal [7]. The orthonasal olfactory perception, by way of odors that reach the olfactory epithelium through the nose even before the food is brought into the mouth, specifically stimulates the appetite for foods containing that aroma [8]. Once the food is inside the mouth, retronasal stimulation decreases the appetite for it and apparently also for other foods. For example, Ramaekers et al. [9] found that the tomato soup intake was reduced by 9% in subjects who perceived the odor of tomato for a longer time and more intensely than in subjects who perceived it less intense and for a shorter time. Furthermore, increasing the perceived intensity of an aroma has been shown to increase the sense of satiety and to reduce food intake [10,11].

The obesity epidemic is a serious health problem associated with an increased risk of mortality and major physical comorbidities such as type 2 diabetes, hypertension, coronary heart disease, stroke and multiple cancer types [12]. Moreover, obesity is frequently associated with mental health disturbances and psychosocial problems, and patients with obesity are exposed to discrimination and stigmatization commonly occurring in different settings with deleterious consequences on mental health [13,14,15]. Although the causal mechanisms and possible therapeutic approaches are extensively studied, in the last 40 years the incidence of overweight and obesity, defined by a body mass index (BMI) between 25–29.9 kg/m^2^ and > 30 kg/m^2^ respectively, has almost tripled, showing a prevalence of 39% for overweight and 13% for obesity [16]. The etiopathogenesis of obesity is complex and multifactorial, including genetic and epigenetic factors, gut microbiota composition, quality of sleep and chronotype [17,18,19,20,21]. However, one of the main factors contributing to the rapid increase in its incidence is the environment in which we live, which encourages the overconsumption of foods rich in energy [22,23,24]. In particular, the overconsumption of foods rich in saturated fats and processed sources of carbohydrates, often associated to a low intake of plant-derived foods rich in fibers, vitamins, and minerals, promotes oxidative stress and low-grade inflammation typical of obesity and related chronic diseases [25]. Stimuli from the surrounding environment, such as food pictures and food odors, have a greater effect on people with obesity than on normal weight individuals [26,27,28,29,30]. Furthermore, individuals with obesity, despite being less sensitive to odors, perceive food odors as more pleasant than normal weight people [31,32].

Based on their olfactory function, individuals can be classified as normosmic (no alteration), hyposmic (reduced perception of odors) or anosmic (general inability to perceive odorants and/or partial for a specific odor) [33,34,35,36]. This variability can be due to both environmental and genetic factors [35,37,38,39,40]. Olfactory deficits are also associated with several chronic diseases such as hypertension, neurodegenerative diseases (Parkinson’s, Alzheimer’s), depression, autoimmune/inflammatory diseases and obesity [41,42,43,44,45,46].

However, the relationship between olfactory performance and obesity is still a matter of debate. Indeed, previous studies have shown that people with obesity have a higher odor detection threshold in both adults [31,47,48] and children [49]. In another study on adult males, a positive correlation was found between subcutaneous adiposity and a lower odor perception [50]. Again, Simchen et al. [51] reported that the relationship between obesity and olfactory abilities is age-dependent: for adults under the age of 65, lower BMIs are associated with better olfactory abilities, while the opposite was found for adult individuals older than 65. Individuals with obesity have been reported to achieve significantly lower scores for odor threshold, discrimination and identification [24,52]. On the contrary, other studies showed no relationship between BMI and olfactory ability, both general and specific for odor discrimination and identification [53,54]. Finally, Stafford et al. reported that people with obesity showed a higher sensitivity to odors than people with lower BMI [32].

Based on these considerations and bearing in mind that people with impaired olfactory function report having changed their eating habits, seeking more appetizing foods and perceiving them as less flavorful and less pleasant [55,56], the aim of our study was to evaluate the olfactory function in patients with overweight and obesity and to compare it with that of healthy normal weight subjects. Since the sense of smell affects food choices and food intake, which in turn contribute to determine the body weight and BMI, we investigated whether the BMI of patients is correlated with olfactory function and whether the olfactory status (normosmic or hyposmic) influences BMI. Finally, by considering that, to the best of our knowledge, most studies have been conducted on women or individuals without taking gender into account, we evaluated in females and males, separately: (a) the effect of the population (patients vs. controls) and of the BMI status (normal weight, overweight and obesity) on the olfactory scores, and (b) the effect of the olfactory status on the BMI and the correlation between olfactory scores and BMI values.

## 2. Materials and Methods 

### 2.1. Subjects

One hundred and thirty-five Caucasian volunteers were recruited in the metropolitan area of Cagliari (Sardinia, Italy) and were divided into two groups. The first group consisted of patients (*n* = 70; 17 men, 53 women; age 54.87 ± 1.76 years), recruited for the study at the Obesity Center of the University Hospital of Cagliari (Sardinia, Italy), hereafter referred to as OC patients, and included subjects with overweight (BMI  25–29.99 kg/m^2^; *n* = 23) and obesity (BMI ≥  30 kg/m^2^; *n* = 47). The second group included healthy control (HC) subjects (*n* = 65; 17 men, 48 women; age 51.11 ± 2.22 years) with normal weight (BMI 18.5–24.99 kg/m^2^) recruited by means of a public announcement at the local university. The two populations were matched for age (*χ*^2^ = 0.6, *p* = 0.80) and gender (*χ*^2^ = 0.01, *p* = 0.94). For both HC subjects and OC patients, exclusion criteria were presence of neurological or psychiatric diseases, pregnancy or lactation, history of cancer, head trauma, sinusitis or nasal sept disorders. Controls and patients who claimed to have had allergic reactions or nasal congestion before undergoing the smell tests were discarded. All subjects were fragrance-free and were required to fast for at least 2 h prior to testing. 

Height, expressed in cm, was measured using a wall-mounted stadiometer (SECA) and body weight expressed in kg was evaluated by means of a calibrated scale (TANITA). The BMI was calculated through the ratio between the weight and the square of height (kg/m^2^), and used to classify the weight status of the subjects. Anthropometric characteristics of participants are shown in Table 1.

The study was conducted in accordance with the Declaration of Helsinki and was approved by the local Ethics Committee. Before being tested, each subject was briefed on the aim of the study and experimental procedures and asked to sign an informed consent.

### 2.2. Olfactory Sensitivity Screening

To evaluate the orthonasal olfactory function of individuals, we adopted the standard “Sniffin’ Sticks” test battery (Burghart Instruments, Wedel, Germany) consisting of three subtests for olfactory threshold (T-test), olfactory discrimination (D-test) and olfactory identification (I-test) [57]. This test is broadly used for olfactory screening, also in the health field, and it is internationally recognized as effective.

For the determination of the olfactory threshold, the experimenter has 48 pens available, divided into 16 triplets, each consisting of two pens containing a solvent and the third soaked in an increasing concentration of n-butanol. The triplets are presented in ascending order of concentration until the subject identifies the pen containing n-butanol twice in a row in the same triplet. This is the point of first inversion, where the triplets start to be presented in decreasing order of dilution of n-butanol. Whenever the subject fails to recognize the target pen, the dilution order of presentation of the triplets is reversed. Upon reaching the seventh inversion the experiment ends and the threshold score is given by the average of the last four inversions. For the determination of odor discrimination, the experimenter has 16 triplets available, each consisting of two pens containing the same odor and one with a different one (target pen). The participant’s goal is to identify the target pen. The score obtained is given by the number of correct answers from 0 to 16. To identify odors, 16 pens are used containing as many odors familiar to the subjects as possible. Each pen is associated with four possibilities for the subject to choose from. In this case, the score is the number of correct identifications from 0 to 16.

During the test, the experimenter compiles a protocol in which the scores obtained for each sub-test, age, gender, height, weight, and BMI are reported. The sum of the scores obtained with the T-test, D-test and I-test gives the total TDI, which is used to classify the subjects according to their general olfactory performance as normosmic or hyposmic. Based on the score obtained with the T-test, D-test and I-test, subjects can also be classified by olfactory threshold, olfactory discrimination and olfactory identification, respectively [58].

### 2.3. Data Analyses 

One-way ANOVA was used to analyze: (a) the effect of the subjects’ health status (HC or OC) on the score obtained with the T-test, D-test, I-test and their TDI sum; (b) the effect of the BMI status (normal weight, overweight and obesity) on the score obtained with the T-test, D-test and I-test; (c) the effect of the olfactory status (normosmic or hyposmic) on the BMI of OC patients.

Two-way ANOVA was used to verify a significant interaction between: (a) gender × health status (HC or OC) and gender × BMI status (normal weight, overweight and obesity) on the score obtained with the T-test, D-test and I-test and their TDI sum; (b) gender × olfactory status (TDI, T, D or I) on BMI presented by OC patients.

Data were checked for the assumptions of homogeneity of variance and normality. Post-hoc comparisons were made with Fisher’s least significant difference (LSD) test or Duncan’s test when homogeneity of variance hypothesis was violated. Statistical analyses were carried out with STATISTICA for WINDOWS (version 7.0; StatSoft Inc., Tulsa, OK, USA). *p* values < 0.05 were considered significant.

Fisher’s exact test was used to analyze differences in the TDI, T, D and I olfactory status between HC subjects and OC patients.

The Pearson’s correlation coefficient was used to evaluate the association between TDI, T, D or I olfactory score vs. BMI, for OC patients, considering females and males both together and separately. Statistical analyses were performed using GraphPad Prism 6 (GraphPad Software, San Diego, CA, USA). *p* values < 0.05 were considered significant.

## 3. Results

### 3.1. Olfactory Scores and Subject Classification

Table 2 shows the distribution of HC subjects and OC patients classified as normosmic or hyposmic based on their overall olfactory function. Fisher’s test revealed a significantly different distribution between controls and patients: in fact, 66% of HC subjects were classified as normosmic, but only 34% of OC patients.

One-way ANOVA showed that the TDI olfactory score obtained by OC patients was significantly lower than that of controls (*F*_1,133_ = 36.2; *p* < 0.001) (Figure 1).

Table 3 shows the distribution of HC subjects and OC patients classified as normosmic or hyposmic based on their ability to perceive (T olfactory status), discriminate (D olfactory status) or identify (I olfactory status) odors. Fisher’s test highlighted a significant difference in the distribution of controls and patients in relation to their D and I olfactory status (*p* < 0.001). In detail, over 95% of the HC subjects were normosmic, while approximately 35% of the OC patients were classified as hyposmic. Finally, no significant differences were found between controls and patients for their T olfactory status.

The mean values ± SEM of the T, D and I olfactory scores obtained by the HC subjects and OC patients are shown in Figure 2. One-way ANOVA revealed a significant effect of patient status vs. control on the D olfactory score (*F*_1,133_ = 28.2; *p* < 0.001) and I olfactory score (*F*_1,133_ = 39.1; *p* < 0.001). On the other hand, no difference was found for the T olfactory score between the two populations (*F*_1,133_ = 3.31; *p* = 0.071).

Figure 3 shows mean values ± SEM of the T, D, and I olfactory scores reached by controls, who were all normal weight subjects, and OC patients divided by their BMI as overweight (OW) or with obesity (Ob). One-way ANOVA showed a significant effect of the BMI status (normal weight, overweight and obesity) on each olfactory score: T (*F*_2,132_ = 8.71; *p* < 0.000), D (*F*_2,132_ = 23.26; *p* < 0.000) and I olfactory score (*F*_2,132_ = 33.82; *p* < 0.000). Post-hoc comparisons indicated that Ob patients achieved T, D and I olfactory scores that were significantly lower than those obtained by both normal weight controls (T: *p* < 0.005, Fisher’s LSD test; D–I: *p* < 0.000, Duncan’s test) and OW patients (T: *p* < 0.000, Fisher’s LSD test; D–I: *p* < 0.000; Duncan’s test). In addition, post-hoc tests showed that I olfactory scores reached by normal weight controls were also higher that those obtained by OW patients (*p* = 0.016; Duncan’s test). These findings indicate that, by order of magnitude, the olfactory scores obtained by subjects according to their BMI were: normal weight > overweight > obesity.

### 3.2. BMI Effects

The mean values ± SEM of BMI determined in OC patients classified by their overall TDI olfactory status and individually by their T, D and I olfactory status are shown in Figure 4. One-way ANOVA revealed that BMI of normosmic patients was significantly lower than that of hyposmic patients (TDI status: *F*_1,68_ = 13.44; *p* < 0.001; T status: *F*_1,68_ = 7.93; *p* = 0.006; D status: *F*_1,68_ = 17.83; *p* < 0.001; I status: *F*_1,68_ = 15.83; *p* < 0.001).

Pearson’s correlation test was used to check for a correlation between BMI and olfactory scores in OC patients (Figure 5). The results indicate that a significant negative correlation exists between the TDI olfactory score obtained by each OC patient and his/her BMI (Pearson’s r = −0.48, *p* < 0.001). The same negative correlations were also found between the BMI of each OC patient and his/her T (Pearson’s r = −0.32, *p* = 0.007), D (Pearson’s r = −0.44, *p* < 0.001) and I (Pearson’s r = −0.37, *p* = 0.001) olfactory scores. The correlations we found, although always significant, are strong between BMI and TDI olfactory score, moderate between BMI and D or I olfactory score, and weak between BMI and T olfactory score.

### 3.3. Olfactory Function and BMI in Women and Men Separately

Figure 6 shows mean values ± SEM of the T, D, I olfactory scores and their summed TDI olfactory score obtained by the HC subjects and OC patients, for females and males separately. Post-hoc comparisons subsequent to two-way ANOVA revealed that both females and males of HC subjects achieved olfactory scores of TDI, D and I which were significantly higher than females and males of OC patients (F: *p* < 0.001; M: *p* ≤ 0.008; Fisher’s LSD test). No difference was found between controls and patients in the T olfactory score, for both females (*p* = 0.119; Fisher’s LSD test subsequent to two-way ANOVA) and males (*p* = 0.324; Fisher’s LSD test).

Figure 7 shows mean values ± SEM of the T, D, I olfactory scores and their summed TDI olfactory score reached by females and males according to their BMI for normal weight HC subjects and OC patients with overweight or obesity. For females, post-hoc tests revealed that patients with obesity obtained T, D, I olfactory scores and their summed TDI score significantly lower than those achieved by both normal weight controls (TDI: *p* < 0.000; T: *p* = 0.002; D: *p* < 0.000; I: *p* < 0.000; Fisher’s LSD test) and overweight patients (TDI: *p* < 0.000; T: *p* < 0.000; D: *p* = 0.003; I: *p* < 0.005; Fisher’s LSD test). Except for the olfactory T score (*p* ≥ 0.434; Fisher’s LSD test), similar results were also found for males, with patients with obesity scoring significantly lower values than both normal-weight controls (TDI: *p* < 0.001; D: *p* < 0.001; I: *p* < 0.001; Fisher’s LSD test) and overweight patients (TDI: *p* = 0.004; D: *p* < 0.044; I: *p* < 0.002; Fisher’s LSD test). No difference was found between normal weight HC and overweight OC patients, except for the I-test score of normal weight females which was significantly higher than that of overweight females (*p* = 0.016; Fisher’s LSD test).

Figure 8 shows the same data according to TDI, T, D and I olfactory status of patients, for females and males separately. Post-hoc comparisons subsequent to two-way ANOVA highlighted that those female patients who were hyposmic showed a higher BMI than normosmic ones for each olfactory status considered (*p* ≤ 0.002; Fisher’s LSD test); male, normosmic patients had a lower BMI than hyposmic ones (*p* ≤ 0.028; Fisher’s LSD test), except for T olfactory status, which although not significantly, was lower in normosmic patients than in hyposmic ones (*p* = 0.167; Fisher’s LSD test).

A negative correlation was also found between BMI and TDI, D and I olfactory scores obtained by each OC patient considering females and males separately (Females: TDI, Pearson’s r = −0.42, *p* = 0.002; D, Pearson’s r = −0.39, *p* = 0.004; I, Pearson’s r = −0.30, *p* = 0.028; Figure 9A) (Males: TDI, Pearson’s r = −0.74, *p* < 0.001; D, Pearson’s r = −0.66, *p* = 0.004; I, Pearson’s r = −0.68, *p* = 0.003; Figure 9B). Instead, for the T olfactory score, a negative correlation was found only for females (Pearson’s r = −0.30, *p* = 0.027), while the statistical analysis did not show any correlation for males (Pearson’s r = −0.48, *p* = 0.052). The correlations between BMI and olfactory scores we found are strong in male patients and between moderate and weak in female patients, although always significant.

## 4. Discussion

The sense of smell plays an important role in the choice of food and can influence the eating behavior of individuals [5]. In fact, olfactory stimuli participate in the modulation of a meal size and can, in some cases, lead to overeating, with consequent weight gain [3,59,60]. Given the relationship between smell and body weight [24], the first objective of the present study was to evaluate the olfactory function in a sample of patients affected by overweight or obesity and compare it with that of a group of normal-weight individuals, who did not differ by age or gender. The results show that OC patients exhibit a reduced olfactory function compared to controls, as evidenced by the olfactory scores obtained in both the odor discrimination and odor identification tests and in the total TDI olfactory score. Furthermore, we found that the number of OC patients classified as hyposmic is significantly higher than the number of HC hyposmic subjects, suggesting that both the general and specific olfactory performance for odor discrimination and identification skills are more impaired in patients than in controls. Likewise, it was previously found that participants with obesity, compared to normal weight controls, have a reduced ability to discriminate odors [61], that a lower BMI is associated with a better ability to identify odors [51], and that people with obesity achieve lower TDI olfactory scores [52]. Instead, we did not find a significantly reduced ability in the olfactory threshold, although we did observe a decline in olfactory sensitivity in patients compared to controls. However, this finding is only partially in contrast with previous studies [31,52]. In fact, when OC patients were divided according to their BMI into overweight (BMI ≥ 25 < 30) and with obesity (BMI > 30) individuals, we found that patients with BMI > 30 scored significantly lower scores than normal weight controls also for the olfactory threshold. Interestingly, on the one hand, the reduced ability to discriminate and identify odors was confirmed for the patients with obesity compared to normal weight controls, while on the other, the scores of the patients with obesity were lower even than those of the patients with overweight, who did not show any difference from normal weight controls. While the sense of smell influences the metabolic state, participating in the cephalic phase of digestive physiology, its function is also modulated by molecules connected to the food intake and by the circulating levels of peptides that regulate energy metabolism [3,59,60]. In particular, orexigenic peptides such as ghrelin increase olfactory sensitivity, which is decreased by the anorexigenic ones such as leptin and insulin [3,62]. We can hypothesize that in the case of individuals with obesity, the role played by these peptides on the olfactory function is two-fold: (1) individuals with obesity are characterized by high circulating levels of leptin and insulin [3] which bind to their particularly abundant receptors in the olfactory structures, thus exerting an inhibitory action mainly on the mitral cells of the olfactory bulb, the first point of olfactory information processing that plays a key role in the representation of the identity and intensity of the smell [63,64,65,66]; (2) in addition, obesity involves a reduction in the circulating levels of ghrelin, with a consequent reduction in the olfactory function due to the reduced effect of this hormone [67]. Taken together, these antagonistic actions may explain the decrease in olfactory function observed in patients with obesity compared to normal weight controls.

Whether olfactory dysfunction can contribute to obesity is still a controversial topic [3,68,69], so the second aim of the study was to evaluate the role of smell on body weight, by evaluating the BMI. The results we found show that patients classified as normosmic have significantly lower BMI than those classified as hyposmic. Besides, we found an inverse correlation between the patients’ BMI and the olfactory score that each of them achieved in the threshold, discrimination and identification tests and in the TDI total olfactory score. Previous studies have shown that olfactory dysfunction impacts food intake, possibly leading individuals to an unbalanced diet [55,56,70,71]. These subjects report that food is less enjoyable and less flavorful and they compensate for the reduced gratification that comes from less olfactory stimulation by preferring more palatable foods, such as sweet and high-fat foods, over fruits and vegetables, and increasing the use of condiments and spices [72,73,74]. Based on these results, it could be hypothesized that normosmic patients have lower BMI values than hyposmic patients because, not having to compensate for the reduced gratification linked to olfactory decompensation, they seek less palatable and energy-rich foods. Furthermore, the sense of smell participates in the responses of the cephalic phase, which in addition to preparing individuals to digest, absorb and metabolize food, intervenes in the processes of beginning and ending a meal [7]. A reduced perception of odors seems to lead to a reduced response of the cephalic phase with consequent delay in reaching satiety and therefore prolonging meals [75,76]. This could lead to a greater intake of high-energy foods and drinks which would explain the contribution of olfactory dysfunction to the modification of eating behavior and obesity [3].

Most studies on the relationship between smell and obesity were conducted in females or individuals regardless of gender [31,55,56,59,60,77]. Therefore the last goal of this study was to evaluate any possible gender-related difference in the relationship between olfaction and obesity. Taken together, the results show that both female and male OC patients exhibit a reduced olfactory function and ability to discriminate and identify odors if compared to female and male HC subjects. It has been suggested that some neural processes, such as odor identification, are required to evaluate the calorie content of foods [60]. Furthermore, an association was found between odor identification and discrimination abilities with diets rich in sugars and saturated fatty acids [78]. These higher-order olfactory skills, which require cognitive functions and semantic memory, appear to be responsible for regulating odor-driven behavior [24]. When patients were divided according to their BMI status into overweight and with obesity, gender-related differences were observed. First, women with obesity showed a higher olfactory threshold, as evidenced by the lower threshold test score, while no difference was found in men. This result could explain why when patients are not gendered, the threshold scores are not significantly different from controls, despite the observed downward trend. Second, again in the case of women, we found that patients who are overweight obtain scores on the I-test that are significantly lower than those of normal weight controls, but still higher than those of patients with obesity. The same gender-related differences are found when female and male OC patients are divided into normosmic and hyposmic. Patients of both sexes with normal olfactory function have a lower BMI than their hyposmic counterparts. Finally, we observed an inverse correlation between BMI and olfactory status, except for BMI and T olfactory status in men, where the significance is borderline (*p* = 0.052).

## 5. Conclusions

In conclusion, our results show that olfactory dysfunction increases with increasing obesity. This may be, at least in part, determined by the metabolic changes observed with an increase in body weight. However, the opposite cannot ruled out; that is, obesity may be, at least in part, caused by olfactory dysfunction. In fact, normosmia seems to protect against excessive body weight gain, as also evidenced by the inverse correlation between BMI and olfactory scores. This could involve the responses of the cephalic phase, with a delay in reaching satiety and an excessive intake of high-energy foods and drinks, thus contributing to obesity. In addition, this study shows that the effects of obesity are the same on the olfactory function of both women and men, and that no gender-related differences were found in the relationship between olfaction and BMI. Therefore, obesity, determined by complex and multifactorial causes and characterized by metabolic imbalances, negatively modifies the olfactory function. Further studies will be needed to better understand which mechanisms and factors (physiological, genetic and environmental) are involved in the complex relationships between olfactory function and obesity, also in relation to gender.

## Figures and Tables

**Figure 1 nutrients-14-01262-f001:**
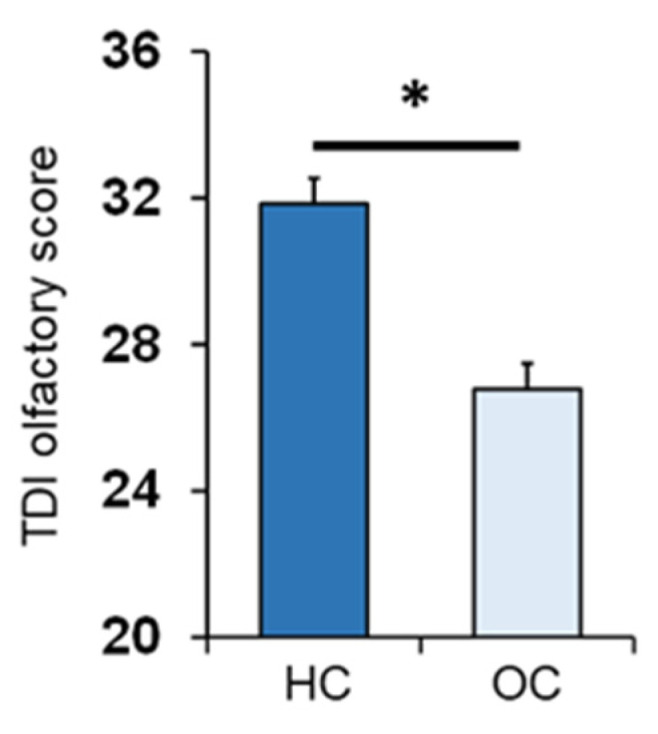
Mean (± SEM) values of the threshold, discrimination and identification (TDI) olfactory score obtained by HC subjects (HC; *n* = 65) and OC patients (OC; *n* = 70). Asterisk indicates significant differences (*p* < 0.001; Fisher’s least significant difference (LSD) test subsequent to one-way ANOVA), HC: healthy control, OC: Obesity Center.

**Figure 2 nutrients-14-01262-f002:**
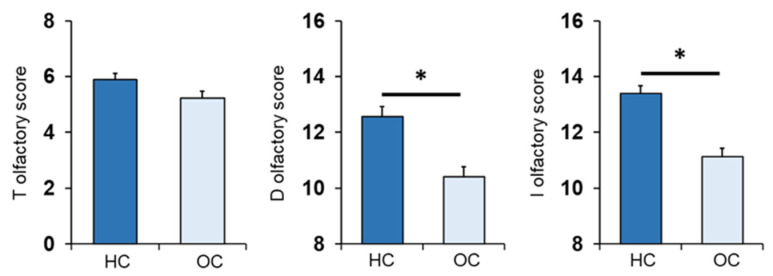
Mean (± SEM) values of the T, D and I olfactory scores determined in HC subjects (*n* = 65), and OC patients (*n* = 70). Asterisks indicate significant differences (*p* < 0.001, Fisher’s LSD test subsequent one-way ANOVA).

**Figure 3 nutrients-14-01262-f003:**
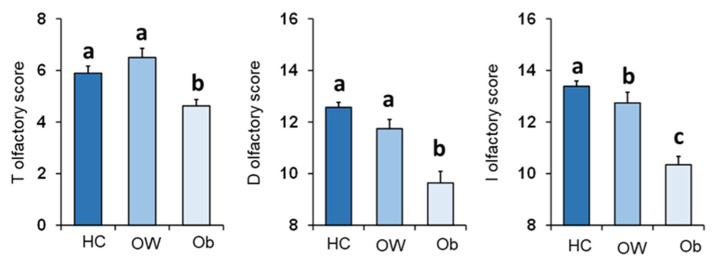
Mean (± SEM) values of the T, D and I olfactory scores determined in HC subjects (*n* = 65), and OC patients divided into overweight (OW; *n* = 23) and with obesity (Ob; *n* = 47). Different letters indicate significant differences (T olfactory score *p* < 0.005, Fisher’s LSD test subsequent to one-way ANOVA; D olfactory score *p* < 0.001; I olfactory score *p* < 0.02; Duncan’s test subsequent to one-way ANOVA).

**Figure 4 nutrients-14-01262-f004:**
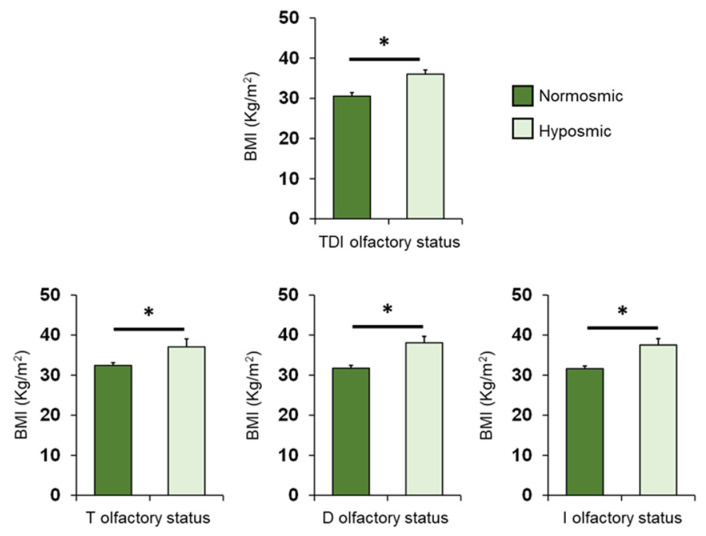
Mean (± SEM) values of BMI determined in OC patients (*n* = 70) classified as normosmic or hyposmic according to their overall TDI olfactory status (normosmic = 28; hyposmic = 42) and separately for their T (normosmic = 49; hyposmic = 21), D (normosmic = 47; hyposmic = 23) and I (normosmic = 44; hyposmic = 26) olfactory status. Asterisks indicate significant differences (TDI: *p* < 0.001; T: *p* = 0.006; D: *p* < 0.001; I: *p* < 0.001; Fisher’s LSD test subsequent to one-way ANOVA).

**Figure 5 nutrients-14-01262-f005:**
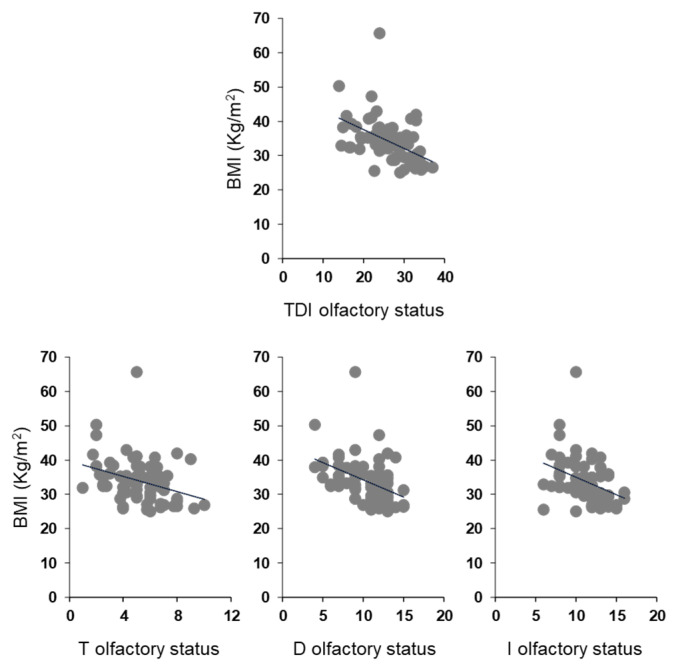
Correlation analysis between BMI and olfactory scores obtained by each OC patient.

**Figure 6 nutrients-14-01262-f006:**
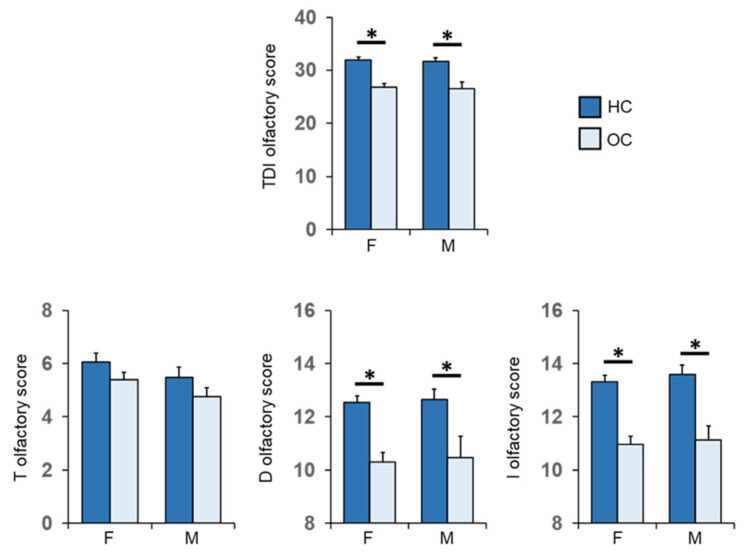
Mean (± SEM) values of the T, D and I olfactory score and their sum TDI olfactory score obtained by HC subjects and OC patients divided into females (F-HC = 48, F-OC = 53) and males (M-HC = 17, M-OC = 17). Asterisk indicates significant differences (F: TDI, D and I, *p* < 0.000; Fisher’s LSD test subsequent to two-way ANOVA) (M: TDI, *p* = 0.002; D, *p* = 0.008; I, *p* < 0.000; Fisher’s LSD test subsequent to two-way ANOVA).

**Figure 7 nutrients-14-01262-f007:**
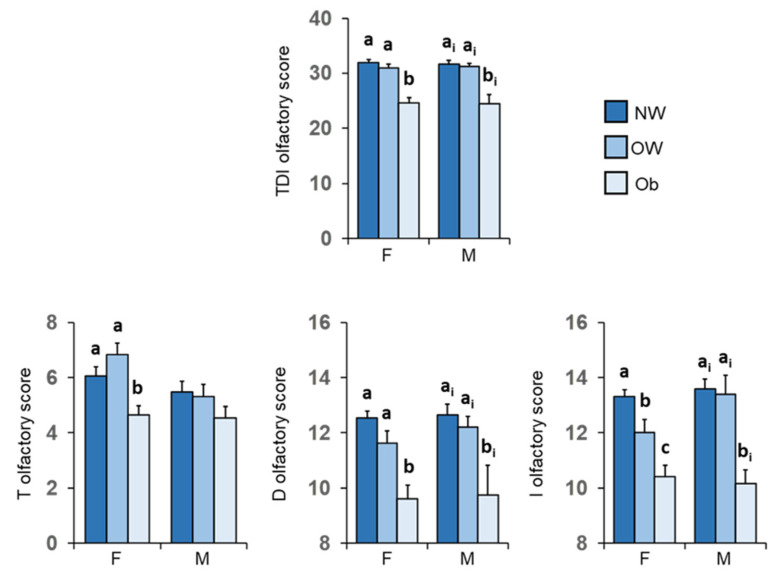
Mean value ± SEM of the T, D and I olfactory scores and their summed TDI score obtained by females and males according to their BMI: normal weight controls (F = 48; M = 17), patients with overweight (F = 18; M = 5) and obesity (F = 35; M = 12). Different letters indicate significant differences (a–c for F: *p* ≤ 0.016; Fisher’s LSD test subsequent to two-way ANOVA) (a_i_–b_i_ for M: *p* ≤ 0.044; Fisher’s LSD test subsequent to two-way ANOVA).

**Figure 8 nutrients-14-01262-f008:**
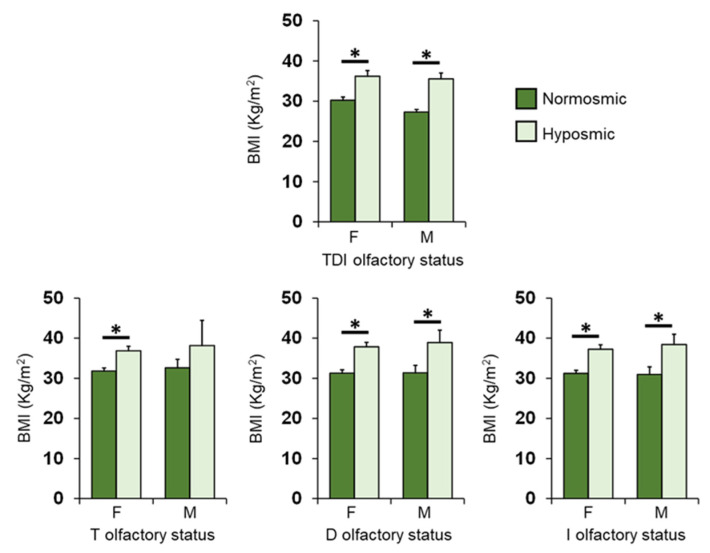
Mean (± SEM) values of BMI determined in female (F) and male (M) OC patients classified as normosmic or hyposmic according to their overall TDI olfactory status (F: normosmic = 24, hyposmic = 29; M: normosmic = 4, hyposmic = 13) and separately for their T (F: normosmic = 35, hyposmic = 18; M: normosmic = 14, hyposmic = 3), D (F: normosmic = 35, hyposmic = 18; M: normosmic = 12, hyposmic = 5) and I (F: normosmic = 32, hyposmic = 21; M: normosmic = 11, hyposmic = 6) olfactory status. Asterisks indicate significant differences (TDI: *p* < 001; T: *p* = 0.006; D: *p* < 0.001; I: *p* < 0.001; Fisher’s LSD test subsequent to one-way ANOVA).

**Figure 9 nutrients-14-01262-f009:**
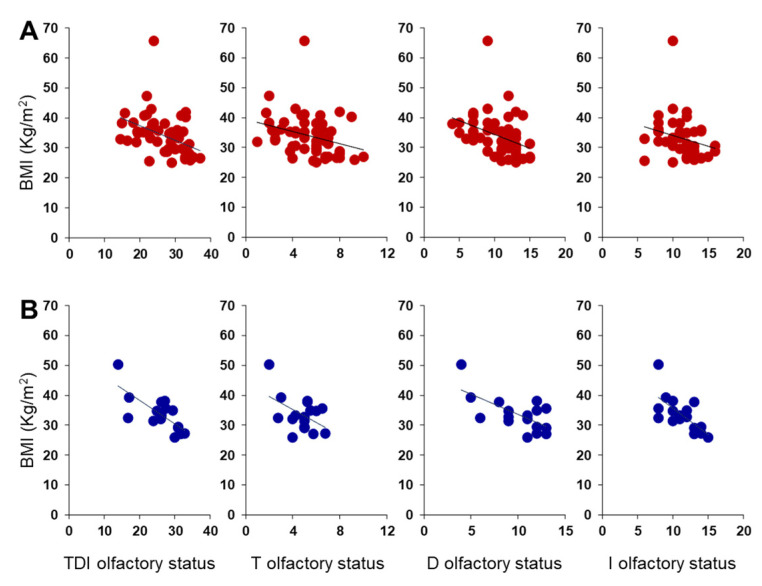
Correlation analysis between BMI and olfactory scores obtained by each OC female (**A**) and male (**B**) patient.

**Table 1 nutrients-14-01262-t001:** Anthropometric characteristics of healthy controls subjects (HC) and Obesity Center (OC) patients.

	HC		OC	
Gender	F	M	F	M
N	48	17	53	17
Age (years)	29–87	26–83	20–81	18–70
Height (m)	1.48–1.80	1.55–1.86	1.38–1.85	1.45–1.85
Weight (Kg)	45–71	54–85.4	50–170	67.8–147.2
BMI (Kg/m^2^)	18.56–24.97	20.06–24.96	25.10–65.6	25.83–50.3

BMI: body mass index.

**Table 2 nutrients-14-01262-t002:** Distribution of healthy control subjects (HC) and Obesity Center (OC) patients classified as normosmic or hyposmic based on their TDI olfactory status.

	Population	HC	OC	*p*-Value
Variable	Olfactory status	*n* (%)	*n* (%)	
TDI	Normosmic	43 (66)	28 (40)	0.002
	Hyposmic	22 (34)	42 (60)	

*p*-value derived from Fisher’s exact test. HC subjects (*n* = 65), OC patients (*n* = 70), TDI: threshold, discrimination and identification.

**Table 3 nutrients-14-01262-t003:** Distribution of the healthy controls subjects (HC) and Obesity Center (OC) patients classified as normosmic or hyposmic based on their Threshold (T), Discrimination (D) and Identification (I) olfactory status.

	Population	HC	OC	*p*-Value
Variable	Olfactory status	*n* (%)	*n* (%)	
T	Normosmic	44 (68)	49 (70)	0.772
	Hyposmic	21 (34)	21 (30)	
D	Normosmic	64 (98)	47 (67)	< 0.001
	Hyposmic	1 (2)	23 (33)	
I	Normosmic	62 (95)	44 (63)	< 0.001
	Hyposmic	3 (5)	26 (37)	

*p*-value derived from Fisher’s exact test. HC subjects (*n* = 65), OC patients (*n* = 70).

## Data Availability

The data presented in this study are available on request from the corresponding author. The data are not publicly available due to restrictions (e.g., privacy or ethical).

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
