# Peer review of "Relationship between Olfactory Function and BMI in Normal Weight Healthy Subjects and Patients with Overweight or Obesity"

_nutrients, 2022, doi:10.3390/nu14061262_

Round 1

Reviewer 1 Report

The manuscript entitled „Relationship between olfactory function and BMI in normal weight healthy subjects and overweight or obese patients” describes an interesting study aimed to evaluate if BMI is associated with impaired olfactory perception.

In my opinion, this topic is super interesting and up-to-date. The study was conducted in a relatively large group, so some conclusions can be made.

However, to improve the quality of the manuscript I would suggest some improvement.

First of all. Please use the person first language! Many associations (including EASO – European Association for the Study of Obesity) raised this issue which is related to the stigma of patients. Thus please change it throughout the text, including the title.

The language requires improvement by language editing service or native speaker. E.g. the first sentence of the abstract makes no sense.

Line 65: due to incorrect translation (I guess), this line is incorrect. Fat is not energy. It is a simplification.

Line 126: please provide the number

Please provide the table with the anthropometric characteristics of participants.

What is SE (mean +/- SE). Standards deviation is always given as SD and other known variety measures are SEM or quartiles.

Figure 3 (and text referring to it). How many subjects were in each group?

Correlations approx. 0.4 are moderate while ~0.3 are weak. Please include it in your description.

Line 404: this statement is too far. The causation is not explained here. Maybe impairment in the olfactory perception is caused by obesity or the opposite. This study did not explain that, so please rewrite the conclusions.

Author Response

Reviewer #1

The manuscript entitled “Relationship between olfactory function and BMI in normal weight healthy subjects and overweight or obese patients” describes an interesting study aimed to evaluate if BMI is associated with impaired olfactory perception.

In my opinion, this topic is super interesting and up-to-date. The study was conducted in a relatively large group, so some conclusions can be made.

However, to improve the quality of the manuscript I would suggest some improvement.

  1. First of all. Please use the person first language! Many associations (including EASO – European Association for the Study of Obesity) raised this issue which is related to the stigma of patients. Thus please change it throughout the text, including the title.

Author’s reply: This suggestion has been accepted and the manuscript has been revised accordingly.

The language requires improvement by language editing service or native speaker. E.g. the first sentence of the abstract makes no sense.

Author’s reply: To comply with the Reviewer’s criticism, in the revised version of the manuscript this sentence has been re-written as follows: “Smell plays a critical role in food choice and intake by influencing energy balance and body weight. Malnutrition problems or modified eating behaviors have been associated with olfactory impairment or loss.” (lines 12-14)

Line 65: due to incorrect translation (I guess), this line is incorrect. Fat is not energy. It is a simplification.

Author’s reply: This comment has been accepted and the sentence “such as fats and sugars” has been deleted from the revised version of the manuscript. (line 66)

Line 126: please provide the number

Author’s reply: This information has been included in the item “Institutional Review Board Statement” reported after the “Discussion” section and before the “References” list and in accordance with the requests of the journal during the submission process. (lines 445-447)

Please provide the table with the anthropometric characteristics of participants. What is SE (mean +/- SE). Standards deviation is always given as SD and other known variety measures are SEM or quartiles.

Author’s reply: To comply with the Reviewer’s suggestion, we added a table (the new Table 1) showing the anthropometric characteristics of the participants in the “2.1 Subjects” paragraph of the “Materials and Methods” section. The revised version of the manuscript has been modified as follows: “Anthropometric characteristics of participants are shown in table 1”. (lines 125-126)

In addition, in the “Results” section “SE” has been replaced with “SEM”.

Figure 3 (and text referring to it). How many subjects were in each group?

Author’s reply: This information is now provided in the Legend of Figure 3 (lines 243-244)

Correlations approx. 0.4 are moderate while ~0.3 are weak. Please include it in your description.

Author’s reply: To comply with the Reviewer’s criticism, the following sentences have been added in the “Results” section of the revised version of the manuscript: “The correlations we found, although always significant, are stronger between BMI and TDI olfactory score, moderate between BMI and D or I olfactory score and weak between BMI and T olfactory score.“ (lines 266-268) and “The correlations between BMI and olfactory scores we found are strong in male patients and between moderate and weak in female ones, although always significant.” (lines 329-331)

Line 404: this statement is too far. The causation is not explained here. Maybe impairment in the olfactory perception is caused by obesity or the opposite. This study did not explain that, so please rewrite the conclusions.

Author’s reply: To comply with the Reviewer’s criticism, the “Conclusions” section has been modified as follows: “In conclusion, our results show that olfactory dysfunction increases with increasing obesity. This may be, at least in part, determined by the metabolic changes observed with an increase in body weight. However, the opposite cannot ruled out, that is obesity may be, at least in part, caused by olfactory dysfunction.” (lines 425-428)

Reviewer 2 Report

The search for factors influencing, including those related to sensory feelings, on excessive consumption leading to overweight and obesity seems to be purposeful and may be used in the prevention and treatment of excess body weight.

However, more information should be provided and issues related to the physiological and nutritional consequences of the various components of the TDI olfactory threshold test (separate explanation of the observed relationships for D - discrimination and I - identification olfactory status) should be discussed. It is especially necessary to explain why there was less or no correlation for T olfactory status when comparing people with normal and excessive body weight. Perhaps it is also worth explaining why the correct perception and discrimination of olfactory does not stimulate higher consumption but rather, as the authors suggest, to reduce it?

Additionally, is it possible that the disturbance of the perception of smells is not only one of the causes, but rather one of the consequences of excessive body weight?

Some of the statistical analyzes performed in these work could be replaced by a multivariate regression analysis.

Authors should provide more detailed information on approval by the Ethics Committee (number and name of the entity that issued the consent).

Author Response

The search for factors influencing, including those related to sensory feelings, on excessive consumption leading to overweight and obesity seems to be purposeful and may be used in the prevention and treatment of excess body weight.

However, more information should be provided and issues related to the physiological and nutritional consequences of the various components of the TDI olfactory threshold test (separate explanation of the observed relationships for D - discrimination and I - identification olfactory status) should be discussed.

Author’s reply: To comply with the Reviewer’s suggstion, the following sentence has been added in the “Discussion” section of the manuscript: “It has been suggested that some neural processes, such as odor identification, are needed to assess the calorie content of foods (Poessel et al 2021). Furthermore, an association was found between odor identification and discrimination ability with diets rich in sugars and saturated fatty acids (Stevenson et al 2016). These higher-order olfactory skills, which require cognitive functions and semantic memory, appear to be responsible for regulating odor-driven behavior (Peng et al 2019).” (lines 405-410)

It is especially necessary to explain why there was less or no correlation for T olfactory status when comparing people with normal and excessive body weight.

Author’s reply: The lack of correlation between BMI status and T olfactory score in male patients could depend both on the sample not large enough to detect differences in the olfactory threshold and on genetic factors. It is known that physiological variations of olfactory threshold can be attributed, at least in part, to a polymorphism of the gene encoding for human Odorant Binding Protein (hOBPIIa), whose main role is to bind the odorants (typically lipophilic) and transport them up to olfactory receptors. Bioinformatics data suggested that mutations in the OBPs genes decreased the expression of OBPs in the olfactory epithelium, by reducing the ability to transport odorants across the mucus layer and, consequently, making the odorant/OR binding more difficult. In the future, it will be interesting to evaluate this polymorphism in patients with overweight and obesity.

Perhaps it is also worth explaining why the correct perception and discrimination of olfactory does not stimulate higher consumption but rather, as the authors suggest, to reduce it?

Author’s reply: Olfactory dysfunction controls not only the quantity, but mainly the quality of the foods that are ingested. In particular, it has been found that individuals with a compromised sense of smell prefer foods with a high energy content. Individuals with reduced olfactory abilities shift their eating habits towards foods rich in sugars and saturated fatty acids, to compensate for the reduced gratification they have from reduced olfactory stimulation.

Additionally, is it possible that the disturbance of the perception of smells is not only one of the causes, but rather one of the consequences of excessive body weight?

Author’s reply: We agree with the Reviewer's comment. In fact, on lines 426-427 we report that metabolic changes caused by weight gain, such as leptin and ghrelin levels, can modify olfactory function. Also, we suggest that a reduced olfactory function can lead to eating high-energy and more flavorful foods which, on the other hand, contribute to increase body weight. Ultimately it is known that there is a relationship between smell and body weight, but it is not yet clear what the cause or effect is. In addition, the following sentence has been added in the “Conclusions” section of the revised version of the manuscript: “However, the opposite cannot ruled out, that is obesity may be, at least in part, caused by olfactory dysfunction”. (lines 427-428)

Some of the statistical analyzes performed in these work could be replaced by a multivariate regression analysis.

Author’s reply: To comply with the Reviewer’s suggestion, we performed the statistical analyses by the linear regression test. As shown below, the results do not change from those obtained with the Pearson’s correlations, both when patients are considered all together and when they are divided by gender.

All patients

Females

Males

Authors should provide more detailed information on approval by the Ethics Committee (number and name of the entity that issued the consent).

Author’s reply: This information has been included in the item “Institutional Review Board Statement” reported after the “Discussion” section and before the “References” list and in accordance with the requests of the Journal during the submission process. (lines 445-447)
